# A statistical-dynamical approach for probabilistic prediction of sub-seasonal precipitation anomalies over 17 hydroclimatic regions in China

Yuan LI[1], Zhiyong WU[1*], Zhiwei ZHU[2], Quan J. WANG[3]

[1] College of Hydrology and Water Resources, Hohai University, Nanjing 210098, China

[2] Key Laboratory of Meteorological Disaster, Ministry of Education (KLME)/ Collaborative Innovation Center on Forecast and Evaluation of Meteorological Disasters (CIC-FEMD)/Joint International Research Laboratory of Climate and Environment Change (ILCEC), Nanjing University of Information Science and Technology,

Nanjing 210044, China

[3] Department of Infrastructure Engineering, The University of Melbourne, Parkville 3010, Australia

*Correspondence to*: Zhiyong WU (wzyhhu@gmail.com)

**Abstract.** Skillful and reliable sub-seasonal precipitation forecasts are of great social and economic value. In

this study, we develop a Spatial Temporal Projection based Calibration, Bridging, and Merging (STP-CBaM) method to improve probabilistic sub-seasonal precipitation forecast skill by combining the strengths of both dynamical models and statistical models. The calibration model is established by post-processing ECMWF raw forecasts using the Bayesian Joint Probability (BJP) approach. The bridging models are built using large-scale atmospheric intraseasonal predictors (U200, U850, OLRA, H200, H500, and H850) defined by the

Spatial-Temporal Projection method (STPM). The calibration model and bridging models are then merged through the Bayesian Modeling Averaging (BMA) method. Our results indicate that the forecast skill of calibration model is higher compared to bridging models when the lead time is within 5-10 days. The U200 and OLRA-based bridging models outperform the calibration model in certain months and certain regions. The BMA merged forecasts take advantage of both calibration model and bridging models. The forecast skill is

further improved compared to the calibration model and bridging models, especially at longer lead times. Meanwhile, the BMA merged forecasts also show high reliability for all regions, months, and lead times. These findings demonstrate the great potential of combing dynamical models and statistical models in improving sub-seasonal precipitation forecasts.



## 1. Introduction


Sub-seasonal forecasting (defined as the time range between 2 weeks and 2 months) bridges the gap between short-medium range weather forecasts and seasonal climate prediction (Vitart and Robertson, 2018). Skillful and reliable sub-seasonal precipitation forecasts are highly valuable for water resource management, flood disaster preparedness, and many other climate-sensitive sectors (White et al., 2022). However, it is considered

a difficult time range to generate skilful forecasts. The memory of atmospheric initial conditions is lost compared to short-medium range forecasts, while the variability of lower boundary conditions, such as sea surface temperature, is too short to take effect (Vitart and Robertson, 2018). Statistical models, which use observational relationships between sub-seasonal precipitation and atmospheric intraseasonal oscillations, have been developed in recent years. The spatial-temporal projection model (STPM), which extracts the

coupled patterns of preceding atmospheric intraseasonal oscillations and precipitation, has shown skill in predicting sub-seasonal precipitation. Zhu and Li (2017) constructed STPMs over different climatic regions during the boreal summer monsoon season, and their results indicated that the STPMs could generate skilful forecasts for intraseasonal precipitation patterns with lead time up to 20 days. Our previous study developed a spatial-temporal projection-based Bayesian hierarchical model (STP-BHM) to take the uncertainties in the

relationships between atmospheric intraseasonal oscillations and sub-seasonal precipitation into account (Li et al., 2022). However, statistical models are highly reliant on stationary relationships between predictors and predictand. Seasonal changes in climatological conditions may lead to different relationships between atmospheric intraseasonal oscillations and precipitation. Liu and Lu (2022) suggested that the impacts of boreal summer intraseasonal oscillation (BSISO) on precipitation are different between early and late summers.

Li et al. (2023) found that the long-period BSISO event-affected region and associated precipitation anomalies are different compared to short-period BSISO events.

With a more comprehensive understanding and better representation of potential sources of predictability, there has been much improvement in dynamical models in recent years. Subseasonal-to-Seasonal Prediction Project (S2S) and the Subseasonal Experiment (SubX) project have been established to provide S2S forecasts

from dynamical models. However, the sub-seasonal precipitation forecasts of GCMs are always of low accuracy (De Andrade et al., 2019; Li et al., 2022). The physical equations are simplified, while the small-scale processes, such as convections, cannot be well represented in most GCMs. In addition, insufficient data assimilation schemes, low capacity in capturing dynamic sources, misrepresentation of atmosphere-ocean



interactions and atmosphere-ocean interactions also contribute to the limited forecast skill (Wu et al., 2023;
Zhang et al., 2021). Although post-processing methods have been proposed in recent years, the forecast skill
after post-processing was still limited for lead time beyond 10-14 days (Li et al., 2021).

Despite the low forecast skill of sub-seasonal precipitation, the GCMs show much higher performance in
predicting large-scale circulation patterns. Cui et al. (2021) evaluated the potential of GCMs for predicting
intraseasonal surface air temperature over mid-high-latitude Eurasia. Their results indicated that the upper
limit of the useful forecast skill ranged from ~10 to ~20 days. The BSISO is the predominant variability of the
Asian summer monsoon, and most GCMs exhibit predictability on timescales of above 3 weeks for BSISO
events (Chen and Zhai, 2017; Hsu et al., 2016; Ren et al., 2018). Lee et al. (2015) evaluated the prediction
skill of BSISO indices using six coupled models in the Intraseasonal Variability Hindcast Experiment project
(ISVHE), and their results suggested that skilful BSISO prediction was about 22 days in strong initial conditions.
Shibuya et al. (2021) suggested that the overall useful prediction skill of the BSISO was approximately 24 days
in a global Non-hydrostatic Icosahedral Atmospheric Model (NICAM) with explicit cloud microphysics. Similar
results were also found by Wu et al. (2023), that the ECMWF model showed skilful prediction of BSISO1 index
at 24-day lead time.

Given the strengths and weaknesses of both statistical models and dynamical models, there has been growing
interest in developing hybrid prediction models that combine forecasts from both statistical and dynamical
models (Slater et al., 2023). Schepen et al. (2014) used POAMA (Predictive Ocean Atmosphere Model for
Australia) forecasts of seasonal climate indices as predictors to predict seasonal precipitation over Australia.
Strazzo et al. (2019b) developed a hybrid statistical-dynamical system to predict seasonal temperature and
precipitation over North America. Most previous statistical-dynamical models focus on seasonal predictions.
Much fewer attempts have been made at sub-seasonal timescales. Specq and Batté (2020) proposed a
statistical-dynamical post-processing scheme to improve the quality of sub-seasonal forecasts of weekly
precipitation using Madden-Julian Oscillation (MJO) and El Niño Southern Oscillation (ENSO) indices as
predictors. Wu et al. (2022) established a dynamical-statistical prediction model (DSPM) to improve sub-
seasonal precipitation forecasts. Deep learning models were also proposed to predict sub-seasonal extreme
rainfall events with the GCM predicted large-scale circulation patterns used as predictors (Xie et al., 2023).
Zhu et al. (2023) developed a dynamical-statistical hybrid model using the novel indices of the zonal





displacements of the South Asia high and the western Pacific subtropical high to predict the Meiyu intraseasonal variation. Nevertheless, the relationships between large-scale circulation patterns and sub-seasonal precipitation are of high uncertainty. More sophisticated hybrid models are required to further improve probabilistic sub-seasonal precipitation forecast skill.

The calibration, bridging, and merging (CBaM) method, which employed Bayes-theorem based approaches to take advantage of both dynamical models and statistical models, have been proven be able to generate skillful and reliable seasonal precipitation and temperature forecasts over different regions (Peng et al., 2014; Schepen et al., 2016; Schepen et al., 2014; Strazzo et al., 2019a). In calibration, the Bayesian joint probability (BJP) approach was used to post-process raw precipitation forecasts derived from GCMs. The BJP approach was also used to generate probabilistic forecasts using large-scale circulation patterns as predictors. This was also referred to as bridging. The calibrated forecasts and bridged forecasts were then merged through the Bayesian model averaging (BMA) method (Wang et al., 2012). Most previous studies used the CBaM method to generate seasonal forecasts. However, much less work has been done on sub-seasonal time scales for several reasons. Compared to seasonal forecasts, there are much fewer climate indices that can be used as predictors at sub-seasonal time scales. Moreover, the atmospheric intraseasonal oscillations may have different effects on precipitation anomalies in different months. As a consequence, it is much more difficult to establish bridging models for sub-seasonal precipitation forecasts. In addition, the evolution of intraseasonal variability of precipitation varies in different stages with different periods in different regions (Liu et al., 2020; Zhu and Li, 2017). The effectiveness of calibration models will be greatly affected if seasonality is not considered.

In this study, we develop a Spatial Temporal Projection based Calibration, Bridging, and Merging (STP-CBaM) method to improve probabilistic sub-seasonal precipitation forecast skill by combining the strengths of both dynamical models and statistical models. The ECMWF sub-seasonal precipitation forecasts are calibrated using the BJP approach for each month. The bridging models are then built using large-scale atmospheric intraseasonal predictors defined by the Spatial-Temporal Projection method (STPM). The calibration model and bridging models are merged through the BMA method to generate skillful and reliable sub-seasonal precipitation forecasts. The STP-CBaM method will be applied to predict sub-seasonal precipitation anomalies over each hydroclimatic region during the boreal summer monsoon from May to October. The accuracy and





reliability will be evaluated through a leave-one-year-out cross-validation strategy.

In the following two sections, data and methodology are introduced. The prediction skill and reliability of the STP-CBaM method are provided in Sect. 4. Sect. 5 discusses the forecast skill, limitations, and future work. Key findings are summarized in Sect. 6.

## 2. Data

### 2.1 Precipitation dataset

In this study, China is divided into 17 hydroclimatic regions on the basis of both climate classifications and watershed division standard (Figure 1). The precipitation data is derived from the latest Multi-Source Weighted-Ensemble Precipitation, version 2.8 (MSWEP V2.8) dataset. This dataset covers the period from 1979 to near recent with a spatial resolution of 0.1° × 0.1°.

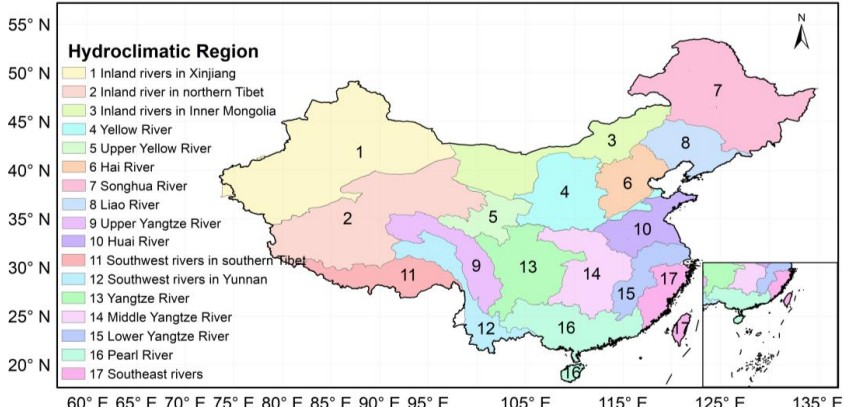

**Figure 1**. The 17 hydroclimatic regions over China.

### 2.2 Reanalysis dataset and OLR dataset

The daily mean geopotential height at 200 hPa, 500 hPa, and 850 hPa (H200, H500, H850), zonal wind at 200 hPa (U200), and 850 hPa (U850), are derived from the ERA5 (Hersbach et al., 2020) reanalysis dataset at https://cds.climate.copernicus.eu/. The daily mean outgoing longwave radiation (OLR) data is provided by the National Oceanic and Atmospheric Administration (NOAA) Physical Sciences Laboratory (PSL), Boulder, Colorado, USA, from their website at https://psl.noaa.gov. The OLR data is developed from high resolution infrared radiation sounder instruments, and has been widely used over the globe. All daily mean data including

U200, U850, OLR, H200, H500, and H850 are bilinear interpolated onto a horizontal resolution of 2.5° × 2.5°

over the period of 2001-2020.

### 2.3 Hindcast dataset

The ECMWF hindcast data of precipitation, U850, U200, OLR, H200, H500, and H850 are retrieved from the

S2S database at http://apps.ecmwf.int/datasets/data/s2s/. Compared to other GCMs, the ECMWF model

shows highest forecast skill in various aspects (Jie et al., 2017; Wu et al., 2023; Zhang et al., 2021). In this

study, we choose hindcasts when the ECMWF model version dates are in the year 2021 from May to October.

Thus, the hindcasts cover the period of 2001-2020. The gridded precipitation hindcasts are area-weighted

averages through 17 hydroclimatic regions as the observational data. In addition, all atmospheric hindcast

fields including U200, U850, OLRA, H200, H500, and H850 are bilinear interpolated onto a horizontal

resolution of 2.5° × 2.5° as the reanalysis dataset.

### 3. Methodology

### 3.1 Intraseasonal signal extraction

In this study, a non-filtering method is used to extract 10-60-day signals for both atmospheric variables (U200,

U850, OLRA, H200, H500, and H850) and precipitation (Hsu et al., 2015; Zhu et al., 2015). The climatological

annual cycle of observational data is first removed by subtracting 90-day low-pass filtered climatological

component. Lower-frequency signals are then removed by subtracting the last 30-day running mean. The

higher-frequency signals are then removed by taking a pentad mean. The so-derived variable represents the

observational 10-60-day signals of daily atmospheric field or precipitation. The daily intraseasonal signals are

then averaged to pentad data to further reduce the noise and improve the predictability. The pentad mean 10-

60-day intraseasonal precipitation is also referred to as pentad mean precipitation anomalies in the following

sections.

As for the hindcast fields of the ECMWF model, the model climatology of atmospheric variables (U200, U850,

OLRA, H200, H500, and H850) and precipitation is removed as a function of initial date and lead time. lower-

frequency signals longer than 60 days are then removed the same way as the observations by subtracting the

running mean of last 30 days. In this process, the observed anomalies before the forecast initial date are used


to make enough data for the running mean. The higher-frequency signals of predicted variables are then removed by taking a pentad mean. The so-derived variable represents the ECMWF model forecasted 10-60-day signals of daily atmospheric field or precipitation.

### 3.2 Model formulation

#### 3.2.1 Predictor definition for bridging models

In this study, we establish the calibration model and bridging models for each hydroclimatic region, month, and lead time. For calibration model, the ensemble mean of ECMWF forecasted pentad mean precipitation anomalies are used as predictors. For bridging models, we define potential predictors using the Spatial-Temporal Projection method (STPM). Relevant areas of atmospheric fields that could affect 10-60-day precipitation variability are found by cell-wise correlation analysis. The effective degree of freedom is estimated following Livezey and Chen (1983).

The spatial-temporal coupled co-variance patterns are constructed for grid points where the correlation is statistically significant at the 5% level. The predictor is then defined by summing the product of the co-variance field derived from the observational data and the ECMWF model forecasted atmospheric intraseasonal signals,

$$cov(X_{i,p}, Y) = \frac{1}{T}\sum_{t=1}^{T}(y_t - E(y))(x_{i,p,t} - E(x_{i,p})) \tag{1}$$

$$X_p^* = \sum_{i=1}^{N} cov(X_{i,p}, Y) * X_{i,p}^* \tag{2}$$

where $X_{i,p}$ denotes the pentad mean 10-60-day signal of $p^{th}$ observational atmospheric field (U200, U850, OLRA, U200, H500, H850) where the correlation is statistically significant at the 5% level for grid $i$ during the training period, $p = 1,2,\cdots,6$. $Y$ denotes the corresponding pentad mean precipitation anomalies. $X_{i,p}^*$ denotes the pentad mean 10-60-day signal of $p^{th}$ hindcast atmospheric field derived from the ECMWF model for grid $i$. $X_p^*$ denotes the $p^{th}$ predictor defined by the STPM method.

#### 3.2.2 Calibration and bridging models

The calibration model and bridging models are established independently from each other, and each model has only one predictor and one predictand. Therefore, there is one calibration model and six bridging models for each hydroclimatic region, month, and lead time.





Each calibration model or bridging model is established using the Bayesian Joint Probability (BJP) approach. The predictor $X_k$, $k = 1, \cdots, K$ and the corresponding predictand $Y$ (pentad mean precipitation anomalies) are normalized to $U_k$ and $V$ using the Yeo-John transformation method,

$$
u_k = \begin{cases}
\frac{\left\{(x_k+1)^{\lambda_{x_k}}-1\right\}}{\lambda_{x_k}} & (x_k \geq 0, \lambda_{x_k} \neq 0) \\
\log(x_k + 1) & (x_k \geq 0, \lambda_{x_k} = 0) \\
-\frac{\left\{(-x_k+1)^{2-\lambda_{x_k}}-1\right\}}{2-\lambda_{x_k}} & (x_k < 0, \lambda_{x_k} \neq 2) \\
\log(-x_k + 1) & (x_k < 0, \lambda_{x_k} = 2)
\end{cases}
\tag{3}
$$

$$
v = \begin{cases}
\frac{\left\{(y+1)^{\lambda_y}-1\right\}}{\lambda_y} & (y \geq 0, \lambda_y \neq 0) \\
\log(y + 1) & (y \geq 0, \lambda_y = 0) \\
-\frac{\left\{(-y+1)^{2-\lambda_y}-1\right\}}{2-\lambda_y} & (y < 0, \lambda_y \neq 2) \\
\log(-y + 1) & (y < 0, \lambda_y = 2)
\end{cases}
\tag{4}
$$

where $\lambda_{x_k}$ and $\lambda_y$ are the unknown transformation parameters for predictor $X_k$ and predictand $Y$.

The matrix $Z^T = [U_k\ V]$ is then assumed to follow a bivariate normal distribution,

$$
Z \sim \mathbf{N}(\boldsymbol{\mu}, \textstyle\sum)
\tag{5}
$$

where $\boldsymbol{\mu}$ and $\sum$ are the mean vector and covariance matrices to be estimated,

$$
\boldsymbol{\mu}^T = [\mu_{U_k}\ \mu_V]
\tag{6}
$$

$$
\textstyle\sum = \boldsymbol{\sigma} R \boldsymbol{\sigma}^T
\tag{7}
$$

where $\boldsymbol{\sigma}$ and $R$ are the standard deviation vector and correlation coefficient matrix, respectively:

$$
\boldsymbol{\sigma}^T = [\sigma_{U_k}\ \sigma_V]
\tag{8}
$$

$$
\mathbf{R} = \begin{bmatrix} 1 & r_{U_k V} \\ r_{V U_k} & 1 \end{bmatrix}
\tag{9}
$$

Note that the correlation coefficient matrix $\mathbf{R}$ is symmetric. Thus, there are only 5 unknown parameters. Here, we denote the 5 unknown parameters of the joint distribution as $\boldsymbol{\theta} = \{\boldsymbol{\mu}, \sum\}$.

Given a data series of $\mathbf{D_0} = \{(x_{k,t}\ y_t),\ t = 1, \cdots, n\}$, We apply the SCE-UA (shuffled complex evolution method developed at The University of Arizona) method to estimate transformation parameters that maximize the log-likelihood function (Duan et al., 1994). The data series of $\mathbf{D_0}$ is then normalized to $\mathbf{D} = \{(u_{k,t}\ v_t),\ t = 1, \cdots, n\}$.

The posterior distribution of $\boldsymbol{\theta}$ is estimated using a Bayesian framework,

$$
p(\boldsymbol{\theta}|\mathbf{D}(U_k, V)) \propto p(\boldsymbol{\theta})p(\mathbf{D}(U_k, V)|\boldsymbol{\theta})
\tag{10}
$$

where $p(\boldsymbol{\theta})$ is the prior distribution of parameters, and $p(\mathbf{D}(U_k, V)|\boldsymbol{\theta})$ is the likelihood. As the posterior distributions of parameters $\boldsymbol{\theta}$ are not standard distributions, analytical integration is difficult. To overcome this problem, we use the new Gibbs sampling algorithm proposed by Wang et al. (2019) to draw a sample of 1000



sets of parameter values. A more detailed description of the sampling strategy can be found in Li et al. (2021).

The posterior predictive distribution of $v_k(t^*)$ is given by

$$f_{V_k}(v_k(t^*)) = \int p(v_k(\mathrm{t}^*)|u_k(t^*),\boldsymbol{\theta})p(\boldsymbol{\theta})p(\mathbf{D}(U_k,V)|\boldsymbol{\theta})\mathrm{d}\boldsymbol{\theta} \qquad (11)$$

where $x_k(t^*)$ is the new forecast value.

Again, the Gibbs Sampling algorithm is used to obtain 1000 samples of $f_{V_k}(v_k(t^*))$. The samples are then

back-transformed to produce the calibrated or bridged predictive density $f_k(y|x_k)$ using parameter $\lambda_y$.

### 3.2.3 Combining models

The merging forecasts are carried out by the Bayesian model averaging (BMA) method proposed by Wang et

al. (2012). Given all candidate models, $f_k(y|x_k)$, $k = 1,\cdots,K$, and the corresponding model weights, $w_k$, $k = $

$1,\cdots,K$, the predictive density of the BMA probabilistic forecasts can be represented as

$$f_{BMA}(y|x_1,\cdots,x_K) = \sum_{i=1}^{K} w_k f_k(y|x_k) \qquad (12)$$

where $x_k$ is the predictor and $y$ is the corresponding predictand.

To encourage even weights among the models, the prior of model weights is assumed to follow a symmetric

Dirichlet distribution, given as

$$p(w_k, k = 1,\cdots,K) \propto \prod_{k=1}^{K}(w_k)^{\alpha-1} \qquad (13)$$

where $\alpha$ is the concentration parameter slightly over 1 and more specifically, $\alpha = 1 + \alpha_0/K$ and $\alpha_0$=0.5. The

posterior distribution of model weights given $t = 1,\cdots,T$ events is as follows:

$$p(w_k, k = 1,\cdots,K|x_k^T,y^T,f_k(y|x_k), k = 1,\cdots,K) \propto \prod_{k=1}^{K}(w_k)^{\alpha-1} \prod_{t=1}^{T}\sum_{k=1}^{K} w_k f_k^{(t)}(y^t|x_k^t) \quad (14)$$

where $f_k^{(t)}(y^t|x_k^t)$ is the cross-validated predictive density. This indicates that the weights are assigned by

the model predictive ability rather than fitting ability. An Expectation-Maximization (EM) algorithm is then used

to estimate the weights by maximizing the likelihood function. Initially, all weights are equal. The EM algorithm

is then iterated until the likelihood function converges.

### 3.3 Evaluation

In this study, a leave-one-year-out cross-validation strategy is used to avoid any bias in skill, including predictor

selection, data normalization, model building, parameter inference, and verification.

The temporal correlation coefficient (TCC) is used to evaluate the performance of the ECMWF model for

predicting atmospheric intraseasonal oscillations. We should note that the ECMWF model has an initial





frequency of twice a week on Tuesday and Thursday. Therefore, 160 or 180 initial dates are found for each

month during the period of 2001-2020. As the atmospheric variables are autocorrelated, the effective degree

of freedom is estimated following Livezey and Chen (1983).

The continuous ranked probability score (Matheson and Winkler, 1976) is used to evaluate the accuracy of

probabilistic forecasts for a given lead time $t$:

$$\text{CRPS} = \frac{1}{N} \sum_{i=1}^{N} \int [F_{i,t}(y) - H(y - o_{i,t})]^2 dy \tag{15}$$

where $F^{i,t}()$ is the cumulative distribution function of the probabilistic forecasts for case $i$ at lead time $t$; and

$H()$ is the Heaviside step function defined as:

$$H(y - o_{i,t}) = \begin{cases} 0 & y < o_{i,t} \\ 1 & y \geq o_{i,t} \end{cases} \tag{16}$$

where $o_{i,t}$ is the corresponding observation.


A CRPS skill score is then calculated by comparing the CRPS of STP-CBaM forecasts with the CRPS of

reference forecasts:

$$\text{CRPS}_{\text{SS}} = \frac{\text{CRPS}_{\text{REF}} - \text{CRPS}}{\text{CRPS}_{\text{REF}}} \times 100\% \tag{17}$$

The reference forecasts are generated using the BJP approach to fit the observations used in the training

dataset. When the CRPS skill score is 100%, the probabilistic forecasts are the same as the observations.

Whereas, a skill score of 0% indicates that the probabilistic forecasts show similar accuracy compared to the

cross-validated climatology. Negative skill score means that the probabilistic forecasts are inferior to the cross-

validated climatology.

The forecast reliability is evaluated using the α-index (Renard et al., 2010). The Probability Integral Transform

(PIT) values of probabilistic forecasts for each case $i$ at lead time $t$ are given as,

$$\pi_{i,t} = F^{i,t}(o_{i,t}) \tag{18}$$

where $F^{i,t}()$ is the cumulative distribution function of probabilistic forecasts, and $o_{i,t}$ is the corresponding

observations. If the ensemble forecasts are reliable, $\pi_{i,t}$ should be uniformly distributed. The $\pi_t$ values are

then summarized into an α-index,

$$\alpha = 1.0 - \frac{2}{N} \sum_{i=1}^{N} \left| \pi_{i,t}^* - \frac{i}{N+1} \right| \tag{19}$$



where $\pi_{i,t}^{*}$ is the sorted $\pi_{i,t}$ in increasing order. The α-index ranges from 0 to 1, and a higher α-index indicates higher reliability.

## 4. Results

### 4.1 Correlation analysis between atmospheric intraseasonal oscillation and precipitation anomalies

Figure 2 presents the correlation between pentad mean 10–60 d signals of U200 and precipitation over Region 1 (inland rivers in Xinjiang) from May to October. The U200 signals near Mongolian Plateau have a positive impact on precipitation anomalies over Region 1 in May, while the impact of U200 signals near eastern Tibetan Plateau is negative. In June and July, the U200 signals in West Siberian Plain and Mongolian Plateau show positive correlations with precipitation anomalies. The spatial patterns of correlations between U200 signals and precipitation anomalies are similar in August, September, and October. The U200 signals near Barents sea and Iranian plateau have positive impacts on precipitation anomalies over Region 1. In comparison, U200 signals over West Siberian Plain show strong negative correlations with precipitation anomalies in these months. The OLRA signals show similar wave patterns as other atmospheric variables (Fig. 3). The spatial patterns of correlations between U850, U200, OLRA, H200, H500, H850 and precipitation anomalies are different for each month as well (Fig. S1~S4).

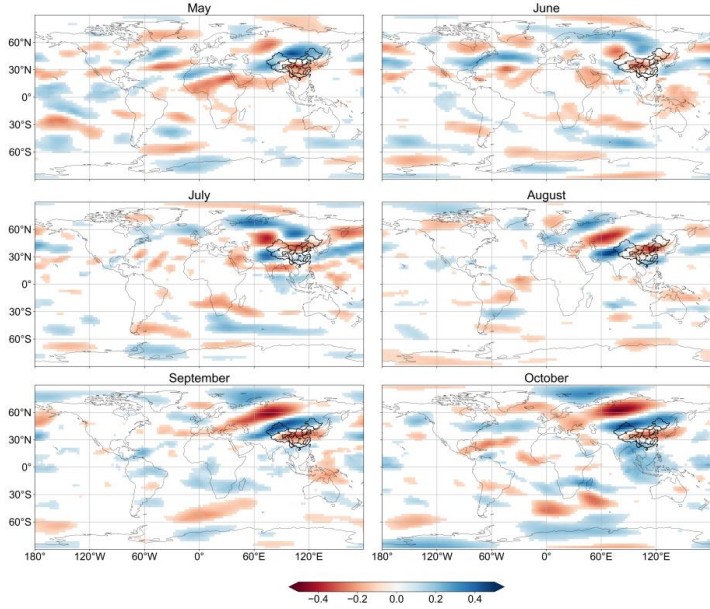

**Figure 2**. Correlation coefficient between pentad mean 10–60 d signals of U200 and precipitation over Region





1 (inland rivers in Xinjiang) in different months. Correlation coefficients that are statistically significant at the

5 % level are shaded.

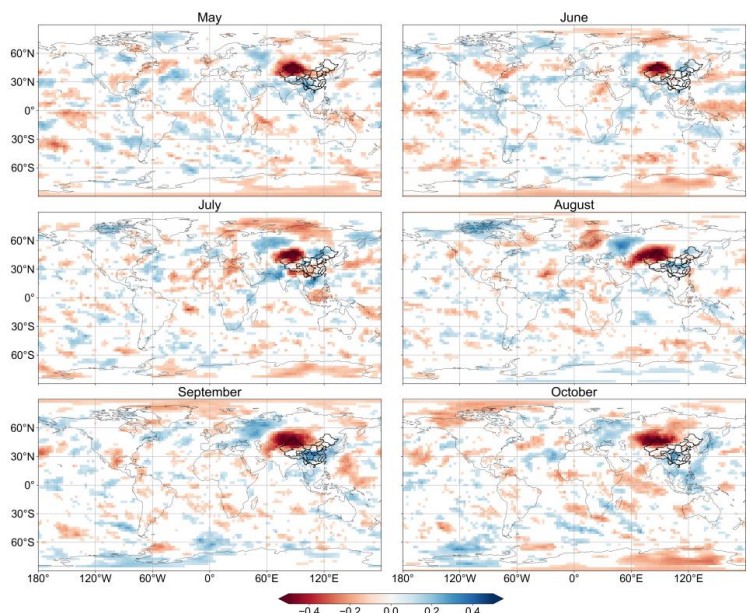

Figure 3. Same as Figure 2, but for OLRA.

**4.2 Skill of ECMWF model in forecasting atmospheric intraseasonal oscillations**

The forecast skill of bridging models is reliant on the forecast skill of atmospheric intraseasonal oscillations

derived from dynamical models. The temporal correlation coefficient (TCC) between the ensemble mean of

ECMWF forecasted U200 intraseasonal signals and the observations in May are shown in Figure 4. The

ECMWF model shows high forecast skill in predicting U200 intraseasonal signals when the lead time is within

10 days, and the correlation coefficients are mostly over 0.7 over the globe. Although the forecast skill

decreases as lead time increases, there are still regions where the forecasted U200 signals are significantly

correlated with the observations. The forecast skill of OLRA intraseasonal oscillations is lower than that of the

U200 signals (Fig. 5). High forecast skill is mostly observed near the equator from 30°N to 30°E when the lead

time is beyond 10 days. Similar results are also found for U850, H200, H500, and H850, where significant

correlations are found mostly near the equator at longer lead times (Fig. S5 to S8). This suggests that the

forecast skill of sub-seasonal precipitation can be potentially improved by taking advantages of both skillful

prediction of atmospheric intraseasonal oscillations and stable relationships between precipitation and large-

scale circulations, especially for tropical regions.

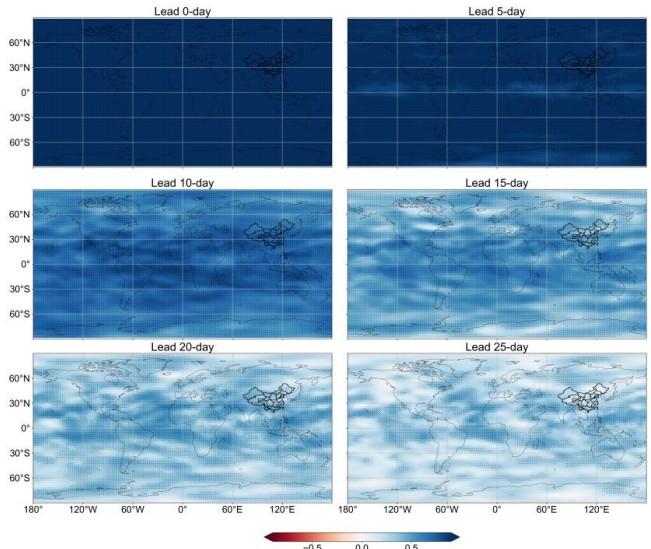

**Figure 4.** Temporal correlation coefficient (TCC) of the ensemble mean of U200 intraseasonal signals derived from the ECMWF model compared to the ERA5 reanalysis data in May. Correlation coefficients that are statistically significant at the 5 % level are shaded.

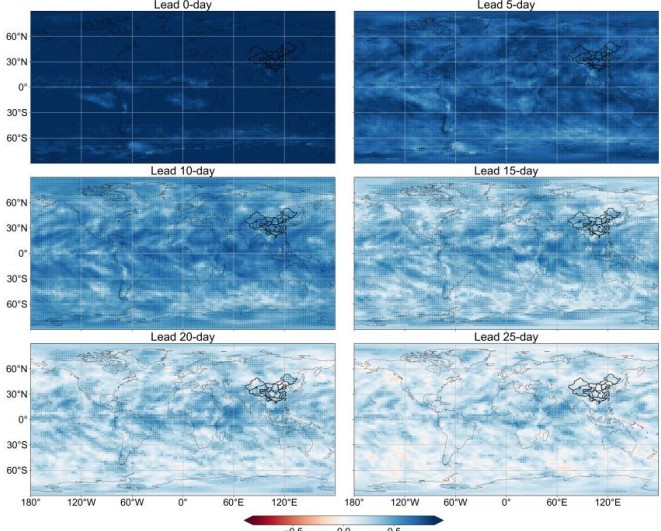

**Figure 5.** Same as Figure 4, but for OLRA.

### 4.3 Skill of calibration model, bridging models, and merged forecasts

Figure 6 presents the spatial distribution of CRPS skill score of calibration model, bridging models, and merged forecasts at different lead times in May. The calibration model shows highest forecast skill compared to bridging

models at short lead times. The forecast skill of calibrated forecasts decreases rapidly, and the CRPS skill scores are mostly below 10% when the lead time is beyond 10 days. The forecast skill of bridging models is higher than the calibration model in Region 10 (Huai River), Region 14 (Middle Yangtze River), and Region 17 (Southeast rivers) at a lead time of 15 days when the OLRA is used as predictor. The forecast skill of bridging

models is higher in Region 6 (Hai River) and Region 7 (Songhua River) when the OLRA and U200 signals are used as predictors at a lead time of 20 days, separately. However, the results also indicate that the accuracy of bridging models are similar to that of the calibration model at longer lead times, partly owing to the relatively lower forecast skill of large-scale circulations at mid-high latitudes. The merged forecasts take advantages of both calibration model and bridging models, and the CRPS skill scores are higher compared to both calibration

model and bridging models especially at longer lead times.

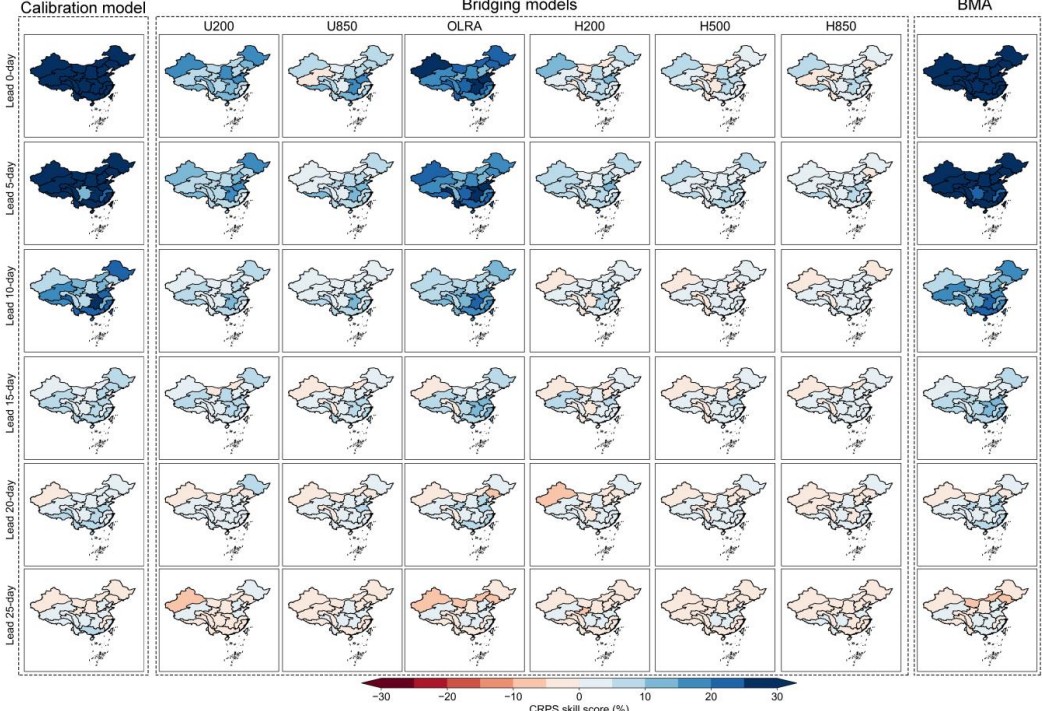

**Figure 6.** CRPS skill score of calibration model, bridging models (U200, U850, OLRA, H200, H500, H850), and merged forecasts (BMA) at different lead times in May.

Figure 7 shows the distribution of model weights at different lead times for Region 1 (Inland rivers in Xinjiang) in May. The weights are rather stable at short lead times, which more than 90% of the total weights are





assigned to the calibration model. Similar results are also found in other regions and months (not shown). The weights of calibration model decrease rapidly when the lead time is beyond 10 days. More weights are assigned to U200 and OLRA at longer lead times. This suggests that the U200 and OLRA signals are more

useful in predicting sub-seasonal precipitation anomalies compared to other large-scale atmospheric circulation variables.

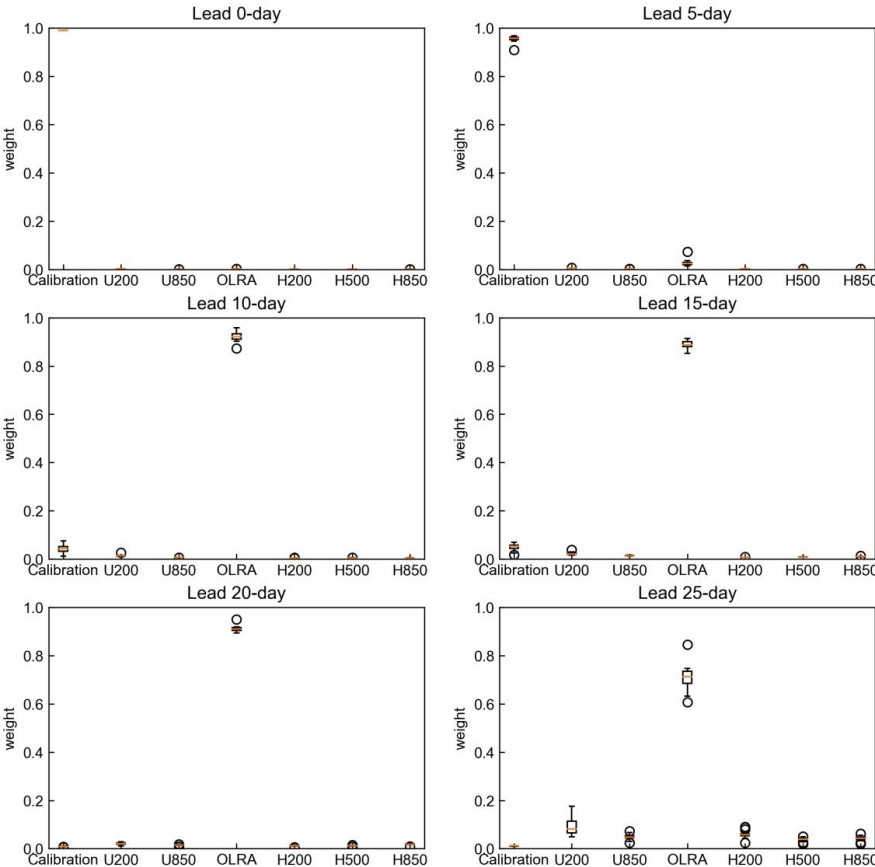

**Figure 7.** Box plots showing the distribution of model weights at different lead times in cross-validation for Region 1 (Inland rivers in Xinjiang) in May.


Figure 8 presents the CRPS skill score of merged forecasts at different lead times from May to October. The forecast skill shows regional, monthly, and lead time-dependent patterns. The CRPS skill scores of merged forecasts are over 30% when the lead time is within 5 days for all regions. Although the forecast skill decreases rapidly as lead time increases, positive skill scores are observed in certain regions and certain months when





the lead time is beyond 15 days. In May, positive CRPS skill scores are found over Region 2, Region 7, Region 9, Region 11, Region 12, Region 14, and Region 15 for all lead times. The spatial patterns of CRPS skill scores are much different in June. The STP-CBaM method outperforms the climatological forecasts in Region 1, Region 4, Region 5, Region 10, Region 13, and Region 16. The STP-CBaM method shows best performance in October, which the CRPS skill scores are positive over most regions for all lead times, except Region 1, Region 7, and Region 8.

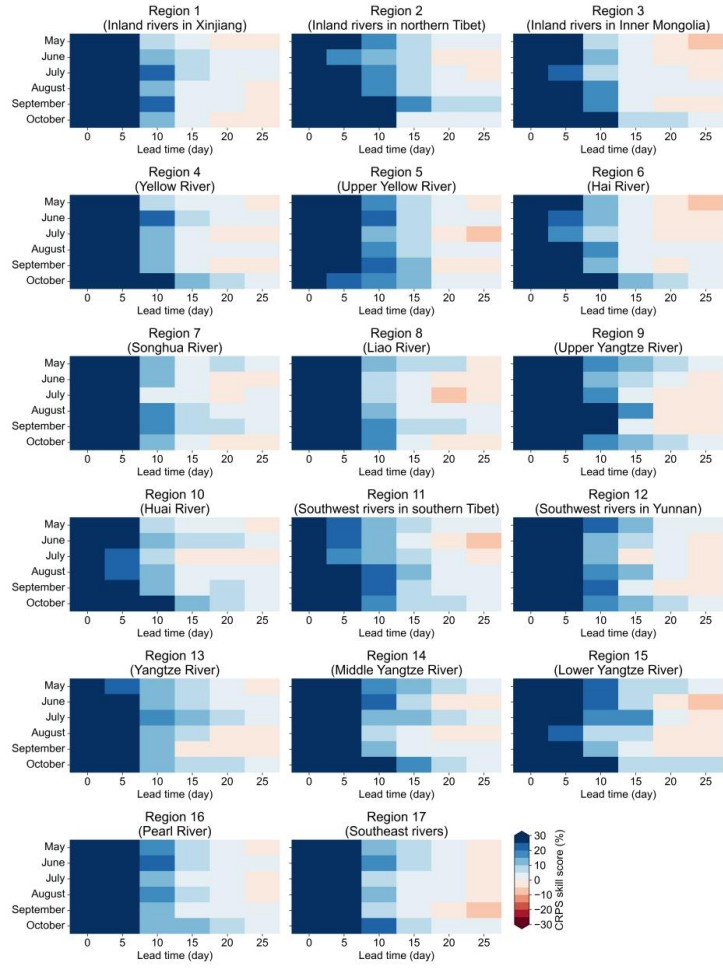

**Figure 8.** CRPS skill score of merged forecasts at different lead times from May to October.

The $\alpha$-index of merged forecasts at different lead times are shown in Figure 9. The values of $\alpha$-index are mostly over 0.7 for all hydroclimatic regions and lead times, suggesting that the merged forecasts are of high





reliability.

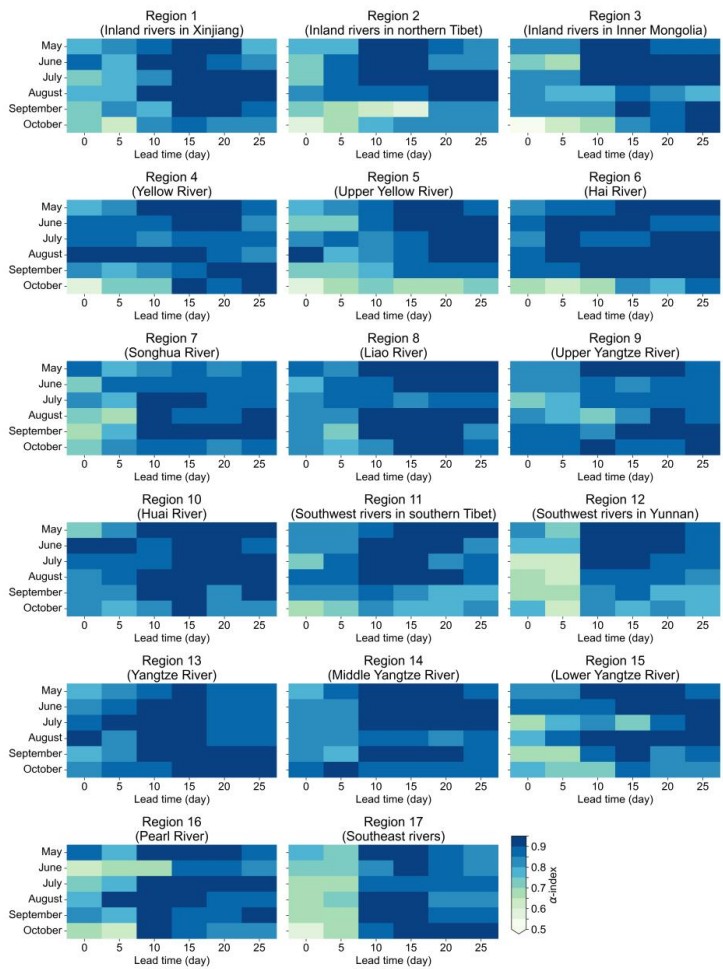

**Figure 9.** Same as Figure 9, but for $\alpha$-index.

## 5. Discussion

### 5.1 Discussion of forecast skill

Through the STP-CBaM model display a good ability to generate skillful and reliable sub-seasonal precipitation forecasts over China, the forecast skill shows great diversity at different regions, months, and lead times. The calibration model shows highest forecast skill compared to bridging models for all regions and all months when the lead time is within 5 days (Fig S9~S13). The U200 and OLRA based bridging models outperform the calibration model and other bridging models when the lead time is beyond 10 days in certain months and

certain regions. This may be explained by the strong relationship between U200, OLRA and precipitation anomalies, and forecast skill of U200 and OLRA in the ECMWF model in these regions (Fig. S5~Fig. S8).

However, we also note that there are several regions where the forecast skill of calibration model is higher than the bridging models at longer lead times. This may be caused by the autocorrelations of sub-seasonal precipitation anomalies defined in this study. In our data processing section, the observed anomalies before the forecast initial date are used to make enough data for the running mean. Thus, the predictand is not purely based on the ECMWF raw forecasts. The observational data is also introduced. The preceding observed precipitation anomalies may provide useful forecast information when the autocorrelations are high. In addition, limited forecast skill of large-scale circulations at mid-high latitudes in dynamical models may contribute to limited forecast skill of bridging models as well.

## 5.2 Limitations and future work

In this study, we aim at investigating the capability of dynamical models for improving the forecast skill of sub-seasonal precipitation anomalies using large-scale circulations as predictors. The bridging models are built based on the concurrent relationships between atmospheric intraseasonal oscillations and precipitation anomalies. Thus, the forecast skill of bridging models is highly reliant on the forecast skill of atmospheric intraseasonal oscillations derived from dynamical models. In the future, the lagged relationships between atmospheric intraseasonal oscillations and precipitation anomalies will be considered to further improve sub-seasonal precipitation forecast skill.

Meanwhile, we define the predictors using the STPM method for each month and each hydroclimatic region. Intraseasonal climate indices, such as the MJO index and BSISO index, have not been considered yet. Recently, Zhu et al. (2023) proposed two sets of novel indices based on the compound zonal displacements of the South Asia high (SAH) and the western Pacific subtropical high (WPH) to monitor and predict the intraseasonal variation of Meiyu. These climate indices will be introduced in the bridging models to improve the forecast skill as well.

In addition, we mainly focus on the prediction of intraseasonal (10-60 day) precipitation anomalies in this study.

400 However, previous studies suggested that the intraseasonal component may only account for 7% of the total variability in northeastern China, while the seasonal component accounted for nearly 70%. Thus, the relationships between seasonal precipitation anomalies and large-scale circulation patterns should also be investigated in these regions in the future.

## 6. Conclusions

405 Skillful and reliable sub-seasonal precipitation forecasts are of high social and economic value. However, it remains a great challenge as the sources of predictability are much less compared to short-medium range forecasts and seasonal forecasts. In this study, we develop a STP-CBaM method to improve probabilistic sub-seasonal precipitation forecast skill by combining the strengths of both dynamical models and statistical models. The calibration model is built by calibrating pentad mean precipitation anomalies derived from the ECMWF

410 model. Potential predictors are defined using the Spatial-Temporal Projection method (STPM) by analyzing the spatial-temporal coupled co-variance patterns between pentad mean precipitation anomalies and large-scale atmospheric circulations (U200, U850, OLRA, H200, H500, and H850). The calibration model and bridging models are merged through the Bayesian Modeling Averaging (BMA) method. The forecast skill and reliability are evaluated through a leave-one-year-out cross-validation strategy.

Our results suggest that the forecast skill of calibration model is higher compared to bridging models when the lead time is within 5-10 days. The U200 and OLRA based bridging models outperform the calibration model when the lead time is beyond 10 days in certain months and certain regions. The BMA merged forecasts take advantage of both calibration model and bridging models. The BMA weights are rather stable at short lead

420 times, which over 90% of the total weights are assigned to the calibration model. More weights are assigned to U200 and OLRA based bridging models at longer lead times. The results of $\alpha$-index suggest that BMA merged forecasts are reliable for all regions, months, and lead times.

In the future, the lagged relationships between atmospheric intraseasonal oscillations and precipitation

425 anomalies will be considered to further improve sub-seasonal precipitation forecast skill. Further improvement may also be achieved by incorporating newly developed climate indices in the bridging models.





**Data availability**

The ERA5 dataset can be sourced from https://cds.climate.copernicus.eu/, and the precipitation dataset is

430 derived from http://www.gloh2o.org/mswep/. The outgoing Longwave radiation (OLR) dataset is sourced at

https://www.ncdc.noaa.gov/news/new-outgoing-longwave-radiation-climate-data-record. The ECMWF

hindcast data can be retrieved from the S2S database at http://apps.ecmwf.int/datasets/data/s2s/.

**Author contribution**

Y.L.: conceptualization, methodology, and writing (original draft preparation, review, and editing). Z.Y. WU:

conceptualization, funding acquisition, supervision, and writing (review and editing). Z.W. ZHU:

conceptualization, review and editing. Q.J. WANG: conceptualization, supervision, and writing (review and

editing).

**Competing interests**

The authors declare that they have no conflict of interest.

**Acknowledgements**

This work was funded by the National Natural Science Foundation of China (Grant numbers: 52009027,
U2240225, and 42088101).

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
