# Peer review of "A statistical-dynamical approach for probabilistic prediction of sub-seasonal precipitation anomalies over 17 hydroclimatic regions in China"

_Hydrology and Earth System Sciences, 2023_

## Author Comment (AC1)

Anonymous Referee #1 Received and published on 23 August 2023.
Our responses are in blue, with the reviewer's comments shown as normal text.

**General comment:**

Skillful and reliable sub-seasonal precipitation forecasts are complicated as the sources of predictability are much fewer than short-medium range and seasonal predictions. This study proposes a Spatial-Temporal Projection based Calibration, Bridging, and Merging (STP-CBaM) method to improve sub-seasonal precipitation forecast skills. The manuscript presents many results demonstrating that the STP-CBaM method can provide skillful and reliable sub-seasonal precipitation forecasts using dynamic and statistical models. The strategy appears novel and therefore merits publication after minor revisions.

Thanks for your comprehensive review and recognition of this study. Your constructive comments will help us improve our manuscript after revision.

**Major comment:**

1. This manuscript uses the intraseasonal oscillation signals forecasted by the ECMWF model as predictors. However, the forecast skill of intraseasonal oscillation is also limited at long lead times. Thus, it is essential to know the potential prediction skill of the STP-CBaM method when observing intraseasonal signals used as predictors.

Thanks for this comment. We agree that it is important to know the potential skill of the STP-CBaM method for sub-seasonal precipitation forecasts. Here, we use the $p^{th}$ 10-60-day signal of atmospheric field derived from ERA5 reanalysis dataset as predictor for the bridging model, instead of the atmospheric field derived from the ECMWF model. Figure S1 presents the potential CRPS skill score of merged forecasts over China. The CRPS skill scores are mostly over 20% even when the lead time is beyond 15 days. Figure S2 shows the differences between potential CRPS skill score and practical CRPS skill score of merged forecasts. The potential CRPS skill scores are slightly lower than the practical CRPS skill scores as the precipitation forecasts derived from the ECMWF model are of high accuracy at short lead times. The potential CRPS skill scores are much higher than the practical CRPS skill scores at longer lead times. This indicates that the forecast skill will be greatly improved when the atmospheric field is well predicted in the GCMs. We will add the above analysis in the revised manuscript.

[Figure]

Figure S1. The potential CRPS skill score of merged forecasts over China

[Figure]

Figure S2. The differences between potential CRPS skill score and practical CRPS skill score of merged forecasts over China

**Minor comments:**

1. Page 1, Line 9,".." is. ".

Thanks for this comment. We will incorporate this suggestion in the revised manuscript.

2. Page 13, The graphical aspect of Figure 4 and Figure 5 could be improved (e.g., colored bars, etc.).

Thanks for this comment. We have revised the color scheme with discrete color for each level. This will make it easier to identify grid points where the correlation coefficients are statistically significant at the 5 % level. Figure 4 and Figure 5 have been revised to improve the visualization as follows:

[Figure]

**Figure 4**. Temporal correlation coefficient (TCC) of the ensemble mean of U200 intraseasonal signals derived from the ECMWF model compared to the ERA5 reanalysis data in May. Correlation coefficients that are statistically significant at the 5 % level are shaded.

[Figure]

**Figure 5**. Same as Figure 4, but for OLRA.

3. Page 14, The CRPS skill scores shown in Figure 6 indicate that the STP-CBaM method can provide more skillful forecasts than the calibration and bridging models alone. However, I noticed that the STP-CBaM forecasts are of lower prediction skill than the calibration model in several regions. The authors should provide some explanations.

Thanks for this comment. We agree that the calibration model outperforms the STP-CBaM model in several regions. To have a better explanation of the results, we compare the CRPS skill of merged forecasts to the CRPS skill score of calibrated forecasts, maximum, mean, and minimum CRPS skill score of bridging forecasts as shown in Figure S3. The CRPS skill scores of merged forecasts lie around the calibrated forecasts at short lead times. Although the CRPS skill scores of merged forecasts are slightly lower than the calibrated forecasts, the CRPS skill scores of merged forecasts are always higher than the minimum CRPS skill scores of bridging forecasts. This indicates that the merged forecasts at least appear to

moderate the worst forecast errors. We will further revise the conclusions to have a more accurate description.

[Figure]

Figure S3. Comparison of the CRPS skill of merged forecasts to the CRPS skill score of calibrated forecasts, maximum, mean, and minimum CRPS skill score of bridging forecasts in May.

4. Page 15, The authors should explain the weights' results in Figure 7.

Thanks for this comment. We noticed that the skill patterns were not always match the weights as suggested by Anonymous Referee #2. At 15-day lead time the weights of OLRA are higher than calibration and U200 in Region 1, but the CRPS of the OLRA is lower than U200 and calibration. In this study, the posterior distributions of model weights are given as

$$p(w_k, k = 1, \cdots, K | x_k^T, y^T, f_k(y|x_k), k = 1, \cdots, K) \propto \prod_{k=1}^{K}(w_k)^{\alpha-1} \prod_{t=1}^{T} \sum_{k=1}^{K} w_k f_k^{(t)}(y^t|x_k^t) \qquad (14)$$

where $f_k^{(t)}(y^t|x_k^t)$ is the cross-validated predictive density.

This indicates that the weights are assigned by the model predictive ability rather than fitting ability. Indeed, there is much literature in support of using predictive performance measures for model choice and combination based on the idea that a model is only as good as its predictions (Eklund and Karlsson, 2007; Stock and Watson, 2006). Thus, the CRPS skill score is not used when inferring model weights. This may lead to the discrepancy between model weights and forecast skill score, especially when none of the models show high predictive skill.

We will have a more detailed discussion on the discrepancy between the model weights and forecast skill in the revised manuscript.

5. L. 359~361: "The values of α-index are mostly over 0.7 for all hydroclimatic regions and lead times, suggesting that the merged forecasts are of high reliability."

I am unsure if the value of the α-index exceeds 0.7, indicating high reliability. I suggest providing some figures to prove such conclusions.

Thanks for this comment. To figure out the differences in reliability between 0.7 and 0.9, we analyze the merged forecasts over Region 3 (Inland Rivers in Inner Mongolia) in May at a lead time of 0-day. The $\alpha$-index of merged forecasts is around 0.6, suggesting that the merged forecasts are of low reliability. We also investigate the model weights of calibrated forecasts and bridging forecasts. The results suggest that the calibrated forecasts are more important than bridging forecasts, which the cross-validated model weights are over 0.95. This suggests that the low reliability of merged forecasts is mostly caused by the low reliability of calibrated forecasts. Figure S4 presents the quantile ranges of calibrated forecasts and merged forecasts against time. The quantile ranges of both calibrated forecasts and merged forecasts are small, suggesting the forecasts are too narrow (too confident). However, we also note that the forecast accuracy of calibrated forecasts is high, which the CRPS skill score is over 60%. We would like to focus on improving the forecast reliability in the future.

[Figure]

[Figure]

Figure S2. Forecast median, quantiles ranges and observed value against time for sub-seasonal forecasts over Region 3 (Inland Rivers in Inner Mongolia) at a lead time of 0-day. The black dots, forecast median; dark blue vertical line, forecast [0.25, 0.75] quantile range; light and dark blue vertical line, forecast [0.10, 0.90] quantile range; red dot, observed precipitation anomalies.

6. L. 365: The authors should also discuss recent progress on the prediction of the East Asian Monsoon. The prediction skill of extreme events should also be addressed.

Liu, B., Zhu, C., Ma, S., Yan, Y., & Jiang, N. (2023). Subseasonal processes of triple extreme heatwaves over the Yangtze River Valley in 2022. Weather and Climate Extremes, 40, 100572.

Yan, Y., Zhu, C., & Liu, B. (2023). Subseasonal predictability of the July 2021 extreme rainfall event over Henan, China, in S2S operational models. Journal of Geophysical Research: Atmospheres, 128(4), e2022JD037879.

Zhu, C., Liu, B., Li, L., Ma, S., Jiang, N., & Yan, Y. (2022). Progress and Prospects of Research on Subseasonal to Seasonal Variability and Prediction of the East Asian Monsoon. Journal of Meteorological Research, 36(5), 677-690.

Thanks for this comment. We will incorporate this suggestion in the introduction section.

Eklund, J. and Karlsson, S.: Forecast Combination and Model Averaging Using Predictive Measures, Econometric Reviews, 26, 329-363, 10.1080/07474930701220550, 2007.

Stock, J. H. and Watson, M. W.: Chapter 10 Forecasting with Many Predictors, in: Handbook of Economic Forecasting, edited by: Elliott, G., Granger, C. W. J., and Timmermann, A., Elsevier, 515-554, https://doi.org/10.1016/S1574-0706(05)01010-4, 2006.

---

## Author Comment (AC2)

Anonymous Referee #2 Received and published on 12 September 2023.

Our responses are in blue, with the reviewer's comments shown as normal text.

**General comment:**

This paper investigates the application of calibration, bridging and merging to forecast subseasonal precipitation anomalies in China based on ECMWF model output of precipitation and atmospheric circulation patterns (zonal winds, geopotential height, OLR). Observations are taken from ERA5, MSWEP and NOAA. Individual models are constructed using BJP and then merged with BMA. Forecast performance is assessed using leave-one-year-out cross-validation and in terms of CRPS, reliability and model weights. It is found that calibration is dominant at short lead times (5-10 days) and U200 and OLRA have increasing relevance at longer lead times. It is concluded that the BMA forecasts have best overall skill and the forecasts are reliable.

Overall, the paper is structured well and the presentation of figures is appropriate. The methods can be followed sufficiently well. My concerns with the paper lie around the marginal performance differences and the strength of the conclusions, particularly around outperformance and reliability. I therefore have some moderate comments for the authors to address ahead of publication, mostly minor, but perhaps requiring some further analysis.

Thanks for your comprehensive review and recognition of this study. Your constructive comments will help us improve our manuscript after revision.

**Major comment:**

Abstract: First line, add a supporting statement about where subseasonal forecasts are of value or delete.

Thanks for this comment. We will remove the first line in the revised manuscript.

Abstract: I suggest adding details about the study area

Thanks for this comment. We will incorporate this suggestion in the revised manuscript.

L37: is it better to say the variability is too slow (rather than too short)?

Thanks for this comment. We will incorporate this suggestion in the revised manuscript.

Section 2.1: Suggest some commentary on the quality of the MSWEP rainfall dataset and underlying sources over the 17 regions.

Thanks for this comment. We will include some commentary on the quality of the MSWEP rainfall dataset as follows:

This dataset is developed by optimally merging precipitation data derived from gauge, satellite, and reanalysis datasets. It covers the period from 1979 to near recent with a spatial resolution of 0.1° × 0.1°. Many studies have found that the MSWEP dataset is of high quality over China (Li et al., 2023b; Liu et al., 2019; Guo et al., 2023).

Li, Y., Pang, B., Zheng, Z., Chen, H., Peng, D., Zhu, Z., and Zuo, D.: Evaluation of Four Satellite Precipitation Products over Mainland China Using Spatial Correlation Analysis, Remote Sensing, 15, 1823, 2023b.

Liu, J., Shangguan, D., Liu, S., Ding, Y., Wang, S., and Wang, X.: Evaluation and comparison of CHIRPS and MSWEP daily-precipitation products in the Qinghai-Tibet Plateau during the period of 1981–2015, Atmospheric Research, 230, 104634, https://doi.org/10.1016/j.atmosres.2019.104634, 2019.

Guo, B., Xu, T., Yang, Q., Zhang, J., Dai, Z., Deng, Y., and Zou, J.: Multiple Spatial and Temporal Scales Evaluation of Eight Satellite Precipitation Products in a Mountainous Catchment of South China, 10.3390/rs15051373, 2023.

Figures 4 & 5: suggest using different colors for the outline

Thanks for this comment. We have revised the color scheme with discrete color for each level. This will make it easier to identify grid points where the correlation coefficients are statistically significant at the 5 % level. Figure 4 and Figure 5 have been revised to improve the visualization as follows:

[Figure]

**Figure 4**. Temporal correlation coefficient (TCC) of the ensemble mean of U200 intraseasonal signals derived from the ECMWF model compared to the ERA5 reanalysis data in May. Correlation coefficients that are statistically significant at the 5 % level are shaded.

[Figure]

**Figure 5**. Same as Figure 4, but for OLRA.

L328-330: It is difficult to interpret the differences visually from blue shading on the maps. To my eye the difference between BMA skill and calibration skill is marginal and sometimes the merged skill is even lower. I suggest the authors find some way to highlight that indeed "the BMA CRPS skill scores are higher compared to both calibration and bridging". Perhaps include some statistics.

Thanks for this comment. We will compare the CRPS skill scores of merged forecasts to the calibrated forecasts, maximum, mean, and minimum CRPS skill score of bridging forecasts as shown in Figure S1. We agree that the CRPS skill scores of merged forecasts are slightly lower than the calibrated forecasts in several regions. Nevertheless, the CRPS skill scores of merged forecasts are always higher than the minimum CRPS skill scores of bridging forecasts. This indicates that the merged forecasts at least appear to moderate the worst forecast errors. We will further revise the conclusions to have a more accurate description.

[Figure]

Figure S1. Comparison of the CRPS skill of merged forecasts to the CRPS skill score of calibrated forecasts, maximum, mean, and minimum CRPS skill score of bridging forecasts in May.

Figure 7: The weights don't seem to match the skill patterns. At 15-day lead time the weights of OLRA are higher than calibration and U200 in Region 1, but the CRPS of the OLRA is lower than U200 and calibration. I suggest the authors discuss the discrepancy.

Thanks for this comment. In this study, the posterior distributions of model weights are given as

$$p(w_k, k = 1, \cdots, K | x_k^T, y^T, f_k(y|x_k), k = 1, \cdots, K) \propto \prod_{k=1}^{K} (w_k)^{\alpha-1} \prod_{t=1}^{T} \sum_{k=1}^{K} w_k f_k^{(t)}(y^t|x_k^t) \qquad (14)$$

where $f_k^{(t)}(y^t|x_k^t)$ is the cross-validated predictive density.

This indicates that the weights are assigned by the model predictive ability rather than fitting ability. Indeed, there is much literature in support of using predictive performance measures for model choice and combination based on the idea that a model is only as good as its predictions (Eklund and Karlsson, 2007; Stock and Watson, 2006). Thus, the CRPS skill score is not used when inferring model weights. This may lead to the discrepancy between model weights and forecast skill score, especially when none of the models show high predictive skill.

We will have a more detailed discussion of the discrepancy in the revised manuscript.

L339-341: Related to the above, it is stated that the OLRA and U200 models are more useful at longer lead times based on the higher weights in Figure 7. However, it is difficult to discern from Figure 6 that the bridging models are skilful beyond about 15 days. I suggest revising this sentence discuss the value in terms of skill rather than the weights alone.

Thanks for this comment. We will incorporate this suggestion in the revised manuscript.

L350-355: I wouldn't give too much credit to weakly positive skill scores at longer lead times, they may not be significantly different from zero. I suggest the paragraph be rewritten to focus on the stronger patterns of skill and not worry too much, e.g., about the differences between May and June at longer lead times.

Thanks for this comment. We will incorporate this suggestion in the revised manuscript.

Figure 9: Some of the reliability index values are quite low, around 0.7, which would indicate some problems with the reliability. I suggest further investigation is required to unpack what is the difference in reliability between 0.7 and 0.9. Perhaps the merging is causing some problems with uncertainty representation.

Thanks for this comment. To figure out the differences in reliability between 0.7 and 0.9, we analyze the merged forecasts over Region 3 (Inland Rivers in Inner Mongolia) in May at a lead time of 0-day. The $\alpha$-index of merged forecasts is around 0.6, suggesting that the merged forecasts are of low reliability. We also investigate the model weights of calibrated forecasts and bridging forecasts. The results suggest that the calibrated forecasts are more important than bridging forecasts, which the cross-validated model weights are over 0.95. This suggests that the low reliability of merged forecasts is mostly caused by the low reliability of calibrated forecasts. Figure S2 presents the quantile ranges of calibrated forecasts and merged forecasts against time. The quantile ranges of both calibrated forecasts and merged forecasts are small, suggesting the forecasts are too narrow (too confident). However, we also note that the forecast accuracy of calibrated

forecasts is high, which the CRPS skill score is over 60%. We would like to focus on improving the forecast reliability in the future.

[Figure]

[Figure]

Figure S2. Forecast median, quantiles ranges and observed value against time for sub-seasonal forecasts over Region 3 (Inland Rivers in Inner Mongolia) at a lead time of 0-day. The black dots, forecast median; dark blue vertical line, forecast [0.25, 0.75] quantile range; light and dark blue vertical line, forecast [0.10, 0.90] quantile range; red dot, observed precipitation anomalies.

Conclusions: I suggest the first and last paragraphs are not really necessary.

Thanks for this comment. We will incorporate this suggestion in the revised manuscript.

L396: It's not certain the skill will be improved, could just say it will be investigated.

Thanks for this comment. We will incorporate this suggestion in the revised manuscript.

Eklund, J. and Karlsson, S.: Forecast Combination and Model Averaging Using Predictive Measures, Econometric Reviews, 26, 329-363, 10.1080/07474930701220550, 2007.

Stock, J. H. and Watson, M. W.: Chapter 10 Forecasting with Many Predictors, in: Handbook of Economic Forecasting, edited by: Elliott, G., Granger, C. W. J., and Timmermann, A., Elsevier, 515-554, https://doi.org/10.1016/S1574-0706(05)01010-4, 2006.

---

## Author Response (AR1)

**Editor decision:**

Dear Authors,

Thank you for your detailed responses to the two reviews. Both reviewers agree the manuscript is of good scientific significance and quality, with very good presentation throughout, and I agree. However, they have both made some valuable suggestions for improving the manuscript. The additional analyses and figures provided in your responses are helpful in addressing these comments. Therefore, I would like to invite you to kindly submit a revised manuscript with these changes incorporated. I look forward to reading the revised manuscript.

Yours,

Louise Slater

We are grateful to you for the kind decision. We have conducted a thorough revision to improve the manuscript as suggested by the insightful and constructive comments of the reviewers. The point-by-point responses are provided in the following.

Responses to Comments on "A statistical-dynamical approach for probabilistic prediction of sub-seasonal precipitation anomalies over 17 hydroclimatic regions in China" (Referee #1)

Anonymous Referee #1 Received and published on 23 August 2023.

Our responses are in blue and the revisions are underlined, with the reviewer's comments shown as normal text.

**General comment:**

Skillful and reliable sub-seasonal precipitation forecasts are complicated as the sources of predictability are much fewer than short-medium range and seasonal predictions. This study proposes a Spatial-Temporal Projection based Calibration, Bridging, and Merging (STP-CBaM) method to improve sub-seasonal precipitation forecast skills. The manuscript presents many results demonstrating that the STP-CBaM method can provide skillful and reliable sub-seasonal precipitation forecasts using dynamic and statistical models. The strategy appears novel and therefore merits publication after minor revisions.

The authors acknowledge the referee's positive comments and the recognition of contribution of this study. We have made thorough revisions to address the comments.

**Major comment:**

1. This manuscript uses the intraseasonal oscillation signals forecasted by the ECMWF model as predictors. However, the forecast skill of intraseasonal oscillation is also limited at long lead times. Thus, it is essential to know the potential prediction skill of the STP-CBaM method when observing intraseasonal signals used as predictors.

Thanks for this comment. We agree that it is important to know the potential skill of the STP-CBaM method for sub-seasonal precipitation forecasts. Here, we use the $p^{th}$ 10-60-day signal of atmospheric field derived from ERA5 reanalysis dataset as predictor for the bridging model, instead of the atmospheric field derived from the ECMWF model. Thus, the potential forecast skill is based on ERA5 reanalysis data, while the practical forecast skill is based on ECMWF model. The potential CRPS skill scores of merged forecasts are then compared to the practical CRPS skill scores in the discussion section from **L. 398** to **L. 403** as follows:

Figure 11 compares the potential CRPS skill score (based on the ERA5 reanalysis) and practical CRPS skill score (based on the ECMWF model) of merged forecasts over China. The potential CRPS skill scores are similar to the practical CRPS skill scores as the precipitation forecasts derived from the ECMWF model are of high accuracy at short lead times. The potential CRPS skill scores are much higher than the practical CRPS skill scores at longer lead times. This indicates that the forecast skill will be greatly improved when the atmospheric field is well predicted in the GCMs.

[Figure]

Figure 11. Practical CRPS skill score of merged forecasts based on the ECMWF model (solid) and potential CRPS skill score based on ERA5 reanalysis (hatched). The red lines are the 50th percentiles, top and bottom of each box are the 75th and 25th percentiles, and whiskers are the maximum and minimum skill scores.

**Minor comments:**

1. Page 1, Line 9,".." is. ".

Thanks for this comment. We have incorporated this suggestion in the revised manuscript.

2. Page 13, The graphical aspect of Figure 4 and Figure 5 could be improved (e.g., colored bars, etc.).

Thanks for this comment. We have revised the color scheme with discrete color for each level. This will make it easier to identify grid points where the correlation coefficients are statistically significant at the 5 % level. Figure 4 and Figure 5 have been revised to improve the visualization as follows:

[Figure]

**Figure 4**. Temporal correlation coefficient (TCC) of the ensemble mean of U200 intraseasonal signals derived from the ECMWF model compared to the ERA5 reanalysis data in May. Correlation coefficients that are statistically significant at the 5 % level are shaded.

[Figure]

**Figure 5**. Same as Figure 4, but for OLRA.

3. Page 14, The CRPS skill scores shown in Figure 6 indicate that the STP-CBaM method can provide more skillful forecasts than the calibration and bridging models alone. However, I noticed that the STP-CBaM forecasts are of lower prediction skill than the calibration model in several regions. The authors should provide some explanations.

Thanks for this comment. We agree that the calibration model outperforms the STP-CBaM model in several regions. To have a better explanation of the results, we compare the distribution of CRPS skill of merged forecasts to the CRPS skill score of calibration model and bridging models as shown in Figure 7. The results are analyzed in the revised manuscript from **L. 327** to **L. 335** as follows:

The merged forecasts take advantages of both calibration model and bridging models. The distribution of CRPS skill score of merged forecasts is similar to the calibration model at a lead time of 0-day. The minimum CRPS skill score of merged forecasts is over 20% in Region 13 (Yangtze River) at a lead time of 5-day, higher than both the calibration model and bridging models. The bridging models, which use the U200 and OLRA as predictors, show higher minimum CRPS skill score compared to the calibration model and other bridging models at a lead time of 10-15 days. The distributions of CRPS skill score of calibration model, bridging models, and the BMA merged forecasts are similar at longer lead times.

[Figure]

Figure 7. Box plots of CRPS skill score of calibration model, bridging models (U200, U850, OLRA, H200, H500, H850), and merged forecasts (BMA) at different lead times in May. The red lines are the 50th percentiles, top and bottom of each box are the 75th and 25th percentiles, and whiskers are the maximum and minimum skill scores.

4. Page 15, The authors should explain the weights' results in Figure 7.

Thanks for this comment. We noticed that the skill patterns were not always match the weights as suggested by Anonymous Referee #2. At 15-day lead time the weights of OLRA are higher than calibration and U200 in Region 1, but the CRPS of the OLRA is lower than U200 and calibration. In this study, the posterior distributions of model weights are given as

$$p(w_k, k = 1, \cdots, K | x_k^T, y^T, f_k(y|x_k), k = 1, \cdots, K) \propto \prod_{k=1}^{K} (w_k)^{\alpha-1} \prod_{t=1}^{T} \sum_{k=1}^{K} w_k f_k^{(t)}(y^t | x_k^t) \tag{14}$$

where $f_k^{(t)}(y^t | x_k^t)$ is the cross-validated predictive density.

This indicates that the weights are assigned by the model predictive ability rather than fitting ability. Indeed, there is much literature in support of using predictive performance measures for model choice and combination based on the idea that a model is only as good as its predictions (Eklund and Karlsson, 2007; Stock and Watson, 2006). Thus, the CRPS skill score is not used when inferring model weights. This may lead to the discrepancy between model weights and forecast skill score, especially when none of the models show high predictive skill.

We have added the above analysis in the revised manuscript from **L. 407** to **L. 412** as follows:

We also note that the weights do not always match the skill patterns. In this study, the posterior distributions of model weights are assigned by the model predictive ability rather than fitting ability. Indeed, there is much literature in support of using predictive performance measures for model choice and combination based on the idea that a model is only as good as its predictions (Stock and Watson, 2006; Eklund and Karlsson, 2007). Thus, the CRPS skill score is not used when inferring model weights. This may lead to the discrepancy between model weights and forecast skill score, especially when none of the models show high predictive skill.

Eklund, J. and Karlsson, S.: Forecast Combination and Model Averaging Using Predictive Measures, Econometric Reviews, 26, 329-363, 10.1080/07474930701220550, 2007.

Stock, J. H. and Watson, M. W.: Chapter 10 Forecasting with Many Predictors, in: Handbook of Economic Forecasting, edited by: Elliott, G., Granger, C. W. J., and Timmermann, A., Elsevier, 515-554, https://doi.org/10.1016/S1574-0706(05)01010-4, 2006.

5. L. 359~361: "The values of α-index are mostly over 0.7 for all hydroclimatic regions and lead times, suggesting that the merged forecasts are of high reliability."
I am unsure if the value of the α-index exceeds 0.7, indicating high reliability. I suggest providing some figures to prove such conclusions.

Thanks for this comment. To figure out the differences in reliability between 0.7 and 0.9, we analyze the merged forecasts over Region 3 (Inland Rivers in Inner Mongolia) in May at a lead time of 0-day. The $\alpha$-index of merged forecasts is around 0.6, suggesting that the merged forecasts are of low reliability. We also investigate the model weights of calibrated forecasts and bridging forecasts. The results suggest that the calibrated forecasts are more important than bridging forecasts, which the cross-validated model weights are over 0.95. This suggests that the low reliability of merged forecasts is mostly caused by the low reliability of calibrated forecasts. Figure S1 presents the quantile ranges of calibrated forecasts and merged forecasts against time (not shown in the manuscript). The quantile ranges of both calibrated forecasts and merged forecasts are small, suggesting the forecasts are too narrow (too confident). However, we also note that the forecast accuracy of calibrated forecasts is high, with the CRPS skill score being over 60%. We would like to focus on improving the forecast reliability in the future.

We have added the above analysis in the revised manuscript from **L. 425** to **L. 427** as follows:
Although the forecast skill of calibration model is high at short lead times, the results also suggest that the calibrated forecasts are too narrow (too confident). We would like to focus on improving the forecast reliability especially at short lead times in the future.

[Figure]

[Figure]

Figure S1. Forecast median, quantile ranges and observed value for sub-seasonal forecasts over Region 3 (Inland Rivers in Inner Mongolia) at a lead time of 0-day. Black dot is forecast median; dark blue vertical line forecast [0.25, 0.75] quantile range; light and dark blue vertical line forecast [0.10, 0.90] quantile range; red dot observed precipitation anomaly.

6. L. 365: The authors should also discuss recent progress on the prediction of the East Asian Monsoon. The prediction skill of extreme events should also be addressed.

Liu, B., Zhu, C., Ma, S., Yan, Y., & Jiang, N. (2023). Subseasonal processes of triple extreme heatwaves over the Yangtze River Valley in 2022. Weather and Climate Extremes, 40, 100572.

Yan, Y., Zhu, C., & Liu, B. (2023). Subseasonal predictability of the July 2021 extreme rainfall event over Henan, China, in S2S operational models. Journal of Geophysical Research: Atmospheres, 128(4), e2022JD037879.

Zhu, C., Liu, B., Li, L., Ma, S., Jiang, N., & Yan, Y. (2022). Progress and Prospects of Research on Subseasonal to Seasonal Variability and Prediction of the East Asian Monsoon. Journal of Meteorological Research, 36(5), 677-690.

Thanks for this comment. We have incorporated this suggestion in the introduction section from **L. 30** to **L. 34** as follows:

Sub-seasonal forecasting (defined as the time range between 2 weeks and 2 months) bridges the gap between short-medium range weather forecasts and seasonal climate prediction (Vitart and Robertson, 2018; **Liu et al., 2023)**. Skillful and reliable sub-seasonal precipitation forecasts are highly valuable for water resource management, flood disaster preparedness, and many other climate-sensitive sectors (White et al., 2022; **Yan et al., 2023; Zhu et al., 2022**).

Responses to Comments on "A statistical-dynamical approach for probabilistic prediction of sub-seasonal precipitation anomalies over 17 hydroclimatic regions in China" (Referee #2)

Anonymous Referee #2 Received and published on 12 September 2023.

Our responses are in blue and the revisions are underlined, with the reviewer's comments shown as normal text.

**General comment:**

This paper investigates the application of calibration, bridging and merging to forecast subseasonal precipitation anomalies in China based on ECMWF model output of precipitation and atmospheric circulation patterns (zonal winds, geopotential height, OLR). Observations are taken from ERA5, MSWEP and NOAA. Individual models are constructed using BJP and then merged with BMA. Forecast performance is assessed using leave-one-year-out cross-validation and in terms of CRPS, reliability and model weights. It is found that calibration is dominant at short lead times (5-10 days) and U200 and OLRA have increasing relevance at longer lead times. It is concluded that the BMA forecasts have best overall skill and the forecasts are reliable.

Overall, the paper is structured well and the presentation of figures is appropriate. The methods can be followed sufficiently well. My concerns with the paper lie around the marginal performance differences and the strength of the conclusions, particularly around outperformance and reliability. I therefore have some moderate comments for the authors to address ahead of publication, mostly minor, but perhaps requiring some further analysis.

Thanks for your comprehensive review and recognition of this study. We have made thorough revisions to address the comments.

**Major comment:**

Abstract: First line, add a supporting statement about where subseasonal forecasts are of value or delete.

Thanks for this comment. We have removed the first line in the revised manuscript.

Abstract: I suggest adding details about the study area

Thanks for this comment. We have incorporated this suggestion in the abstract section.

L37: is it better to say the variability is too slow (rather than too short)?

Thanks for this comment. We have incorporated this suggestion in the revised manuscript in **L. 36**.

Section 2.1: Suggest some commentary on the quality of the MSWEP rainfall dataset and underlying sources over the 17 regions.

Thanks for this comment. We have included some commentary on the quality of the MSWEP rainfall dataset from **L. 128** to **L. 131** as follows:

This dataset is developed by optimally merging precipitation data from gauge, satellite, and reanalysis datasets. It covers the period from 1979 to near recent with a spatial resolution of 0.1° × 0.1°. Many studies have found that the MSWEP dataset is of high quality over China (Li et al., 2023b; Liu et al., 2019; Guo et al., 2023).

Li, Y., Pang, B., Zheng, Z., Chen, H., Peng, D., Zhu, Z., and Zuo, D.: Evaluation of Four Satellite Precipitation Products over Mainland China Using Spatial Correlation Analysis, Remote Sensing, 15, 1823, 2023b.

Liu, J., Shangguan, D., Liu, S., Ding, Y., Wang, S., and Wang, X.: Evaluation and comparison of CHIRPS and MSWEP daily-precipitation products in the Qinghai-Tibet Plateau during the period of 1981–2015, Atmospheric Research, 230, 104634, https://doi.org/10.1016/j.atmosres.2019.104634, 2019.

Guo, B., Xu, T., Yang, Q., Zhang, J., Dai, Z., Deng, Y., and Zou, J.: Multiple Spatial and Temporal Scales Evaluation of Eight Satellite Precipitation Products in a Mountainous Catchment of South China, 10.3390/rs15051373, 2023.

Figures 4 & 5: suggest using different colors for the outline

Thanks for this comment. We have revised the color scheme with discrete color for each level. This will make it easier to identify grid points where the correlation coefficients are statistically significant at the 5 % level. Figure 4 and Figure 5 have been revised to improve the visualization as follows:

[Figure]

**Figure 4**. Temporal correlation coefficient (TCC) of the ensemble mean of U200 intraseasonal signals derived from the ECMWF model compared to the ERA5 reanalysis data in May. Correlation coefficients that are statistically significant at the 5 % level are shaded.

[Figure]

**Figure 5**. Same as Figure 4, but for OLRA.

L328-330: It is difficult to interpret the differences visually from blue shading on the maps. To my eye the difference between BMA skill and calibration skill is marginal and sometimes the merged skill is even lower. I suggest the authors find some way to highlight that indeed "the BMA CRPS skill scores are higher compared to both calibration and bridging". Perhaps include some statistics.

Thanks for this comment. We agree that the difference between BMA skill and calibration skill is marginal especially at longer lead times. To have a better explanation of the results, we compare the distribution of CRPS skill of merged forecasts to the CRPS skill score of calibration model and bridging models as shown in Figure 7. The results are analyzed in the revised manuscript from **L. 327** to **L. 335** as follows:

The merged forecasts take advantages of both calibration model and bridging models. The distribution of CRPS skill score of merged forecasts is similar to the calibration model at a lead time of 0-day. The minimum CRPS skill score of merged forecasts is over 20% in Region 13 (Yangtze River) at a lead time of 5-day, higher than both the calibration model and bridging models. The bridging models, which use the U200 and OLRA as predictors, show higher minimum CRPS skill score compared to the calibration model and other bridging models at a lead time of 10-15 days. The distributions of CRPS skill score of calibration model, bridging models, and the BMA merged forecasts are similar at longer lead times.

[Figure]

Figure 7. Box plots of CRPS skill score of calibration model, bridging models (U200, U850, OLRA, H200, H500, H850), and merged forecasts (BMA) at different lead times in May.

Figure 7: The weights don't seem to match the skill patterns. At 15-day lead time the weights of OLRA are higher than calibration and U200 in Region 1, but the CRPS of the OLRA is lower than U200 and calibration. I suggest the authors discuss the discrepancy.

Thanks for this comment. In this study, the posterior distributions of model weights are given as

$$p(w_k, k = 1, \cdots, K | x_k^T, y^T, f_k(y|x_k), k = 1, \cdots, K) \propto \prod_{k=1}^{K}(w_k)^{\alpha-1} \prod_{t=1}^{T} \sum_{k=1}^{K} w_k f_k^{(t)}(y^t|x_k^t) \qquad (14)$$

where $f_k^{(t)}(y^t|x_k^t)$ is the cross-validated predictive density.

This indicates that the weights are assigned by the model predictive ability rather than fitting ability. Indeed, there is much literature in support of using predictive performance measures for model choice and combination based on the idea that a model is only as good as its predictions (Eklund and Karlsson, 2007; Stock and Watson, 2006). Thus, the CRPS skill score is not used when inferring model weights. This may lead to the discrepancy between model weights and forecast skill score, especially when none of the models show high predictive skill.

We have added the above analysis in the revised manuscript from **L. 407** to **L. 412** as follows:

We also note that the weights are not always match the skill patterns. In this study, the posterior distributions of model weights are assigned by the model predictive ability rather than fitting ability. Indeed, there is much literature in support of using predictive performance measures for model choice and combination based on the idea that a model is only as good as its predictions (Stock and Watson, 2006; Eklund and Karlsson, 2007). Thus, the CRPS skill score is not used when inferring model weights. This may lead to the discrepancy between model weights and forecast skill score, especially when none of the models show high predictive skill.

Eklund, J. and Karlsson, S.: Forecast Combination and Model Averaging Using Predictive Measures, Econometric Reviews, 26, 329-363, 10.1080/07474930701220550, 2007.

Stock, J. H. and Watson, M. W.: Chapter 10 Forecasting with Many Predictors, in: Handbook of Economic Forecasting, edited by: Elliott, G., Granger, C. W. J., and Timmermann, A., Elsevier, 515-554, https://doi.org/10.1016/S1574-0706(05)01010-4, 2006.

L339-341:  Related to the above, it is stated that the OLRA and U200 models are more useful at longer lead times based on the higher weights in Figure 7. However, it is difficult to discern from Figure 6 that the bridging models are skilful beyond about 15 days. I suggest revising this sentence discuss the value in terms of skill rather than the weights alone.

Thanks for this comment. We have revised this sentence from **L.347** to **L. 351** as follows:

The weights of calibration model decrease rapidly when the lead time is beyond 10 days. More weights are assigned to U200 and OLRA at longer lead times. This is mostly consistent with the distribution of CRPS skill scores shown in Figure 7. The CRPS skill scores of U200 and OLRA based bridging models are higher than the calibration model and other bridging models, especially when the lead time is between 10 - 20 days.

L350-355: I wouldn't give too much credit to weakly positive skill scores at longer lead times, they may not be significantly different from zero. I suggest the paragraph be rewritten to focus on the stronger patterns of skill and not worry too much, e.g., about the differences between May and June at longer lead times.

Thanks for this comment. We have revised these sentences to focus on the skill patterns from **L. 358** to **L. 366** as follows:

In general, the forecast skill shows regional, monthly, and lead time-dependent patterns. The merged forecasts show higher skill in predicting sub-seasonal precipitation anomalies over Region 16 (Pearl River) than other regions. The CRPS skill scores are positive for all months at a lead time of 0-20 days. This is mainly owing to the higher prediction skill of OLRA in these regions as shown in Figure 5. In addition, the merged forecasts show highest skill in October, with positive skill scores found over 14 hydroclimatic regions for all lead times except Region 1 (Inland rivers in Xinjiang), Region 7 (Songhua River), and Region

8 (Liao River). In comparison, positive skill scores are found only over 3 hydroclimatic regions for all lead times (Region 1, Inland rivers in Xinjiang, Region 13, Yangtze River, and Region 14, Middle Yangtze River) in July.

Figure 9: Some of the reliability index values are quite low, around 0.7, which would indicate some problems with the reliability. I suggest further investigation is required to unpack what is the difference in reliability between 0.7 and 0.9. Perhaps the merging is causing some problems with uncertainty representation.

Thanks for this comment. To figure out the differences in reliability between 0.7 and 0.9, we analyze the merged forecasts over Region 3 (Inland Rivers in Inner Mongolia) in May at a lead time of 0-day. The $\alpha$-index of merged forecasts is around 0.6, suggesting that the merged forecasts are of low reliability. We also investigate the model weights of calibrated forecasts and bridging forecasts. The results suggest that the calibrated forecasts are more important than bridging forecasts, which the cross-validated model weights are over 0.95. This suggests that the low reliability of merged forecasts is mostly caused by the low reliability of calibrated forecasts. Figure S1 presents the quantile ranges of calibrated forecasts and merged forecasts against time. The quantile ranges of both calibrated forecasts and merged forecasts are small, suggesting the forecasts are too narrow (too confident). However, we also note that the forecast accuracy of calibrated forecasts is high, which the CRPS skill score is over 60%. We would like to focus on improving the forecast reliability in the future.

We have added the above analysis in the revised manuscript from **L. 421** to **L. 423** as follows:

Although the forecast skill of calibration model is high at short lead times, the results also suggest that the calibrated forecasts are too narrow (too confident). We would like to focus on improving the forecast reliability especially at short lead times in the future.

[Figure]

[Figure]

Figure S1. Forecast median, quantiles ranges and observed value against time for sub-seasonal forecasts over Region 3 (Inland Rivers in Inner Mongolia) at a lead time of 0-day. The black dots, forecast median; dark blue vertical line, forecast [0.25, 0.75] quantile range; light and dark blue vertical line, forecast [0.10, 0.90] quantile range; red dot, observed precipitation anomaly.

Conclusions: I suggest the first and last paragraphs are not really necessary.

Thanks for this comment. We have incorporated this suggestion in the revised manuscript from **L. 442** to **L. 454** as follows:

In this study, we develop a STP-CBaM method to improve probabilistic sub-seasonal precipitation forecast skill over 17 hydroclimatic regions in China. The STP-CBaM method takes advantage of both dynamical models and statistical models. The calibration model is built by calibrating pentad mean precipitation

anomalies derived from the ECMWF model. Bridging models are built by defining potential predictors using the Spatial-Temporal Projection method (STPM) method. The calibration model and bridging models are merged through the Bayesian Modeling Averaging (BMA) method. Our results suggest that the forecast skill of calibration model is higher compared to bridging models when the lead time is within 5-10 days. The U200 and OLRA based bridging models outperform the calibration model when the lead time is beyond 10 days in certain months and certain regions. The BMA merged forecasts take advantage of both calibration model and bridging models. The BMA weights are rather stable at short lead times, which over 90% of the total weights are assigned to the calibration model. More weights are assigned to U200 and OLRA based bridging models when the lead time is beyond 10 days. The results of α-index suggest that BMA merged forecasts are reliable, especially at longer lead times.

L396: It's not certain the skill will be improved, could just say it will be investigated.

Thanks for this comment. We have incorporated this suggestion in the revised manuscript from **L. 433** to **L. 434** as follows:

These climate indices will be introduced in the bridging models to investigate the potential improvement for forecast skill.

---

## Author Response (AR2)

**Editor decision:**

Dear Authors,

Thank you for submitting your revised manuscript, which has been read by myself and the same two original reviewers. All agree that the work is of high significance, quality and presentation. One reviewer has made an additional suggestion that is worth addressing, and will be re-reviewed by myself. I look forward to reading your final manuscript.

Yours,

Louise Slater

We are grateful for your kind decision. We have revised the manuscript to address the additional suggestion from Anonymous Referee #2. The responses are provided in the following.

Responses to Comments on "A statistical-dynamical approach for probabilistic prediction of sub-seasonal precipitation anomalies over 17 hydroclimatic regions in China" (Referee #2)

Anonymous Referee #2 Received and published on 2 November 2023.

Our responses are in blue and the revisions are underlined, with the reviewer's comments shown as normal text.

**Suggestions for revision:**

Around L390 states that the relatively low reliability of merged forecasts at short lead times is due to the merged forecasts being too confident (no results shown). Later, around L445 it is discussed that the reliability of calibrated forecasts is poor. These statements require the reader to connect the dots. I suggest adding some text around L390 about the overconfidence of the calibration forecasts being the main cause of the merged forecasts becoming overconfident. It would be useful to refer to a figure in supplementary material, such as the quantile plots included in the response, or perhaps alpha-index plots for calibration & bridging. In this way, the two statements around L390 and L445 are linked and have evidence. Related to this, the abstract should be modified to say that some improvements to reliability are still needed at shorter lead times rather than reliability is high.

Thanks for your valuable suggestion.

We have added several sentences from **L. 374** to **L. 382** to provide evidence for the overconfidence at short lead times as follows:

To figure out the relatively low reliability at short lead times, we analyze the merged forecasts over Region 3 (Inland Rivers in Inner Mongolia) in May at a lead time of 0-day. The α-index of merged forecasts is around 0.6, suggesting that the merged forecasts are of low reliability. We also investigate the model weights of calibrated forecasts and bridging forecasts. The results suggest that the calibrated forecasts are more important than bridging forecasts, which the cross-validated model weights are over 0.95. This suggests that the low reliability of merged forecasts is mostly caused by the low reliability of calibrated forecasts. Figure S9 presents the quantile ranges of calibrated forecasts and merged forecasts against time. The quantile ranges of both calibrated forecasts and merged forecasts are small, suggesting the forecasts are too narrow (too confident).

The quantile plots included in previous response are added as **supplementary material** in the revised manuscript as well.

The abstract has also been revised from **L. 23** to **L. 25** as follows:

Meanwhile, the BMA merged forecasts also show high reliability at longer lead times. However, some improvements to reliability are still needed at shorter lead times.

Typo:

L430: "do not always match"

Thanks for this comment. We have incorporated this suggestion in the revised manuscript.